# Dietary Intake of Rosmarinic Acid Increases Serum Inhibitory Activity in Amyloid A Aggregation and Suppresses Deposition in the Organs of Mice

**DOI:** 10.3390/ijms21176031

**Published:** 2020-08-21

**Authors:** Xuguang Lin, Kenichi Watanabe, Masahiro Kuragano, Yukina Kurotaki, Ushio Nakanishi, Kiyotaka Tokuraku

**Affiliations:** 1Graduate School of Engineering, Muroran Institute of Technology, Muroran 050-8585, Japan; linxuguang4000@163.com (X.L.); gano@mmm.muroran-it.ac.jp (M.K.); 17023039@mmm.muroran-it.ac.jp (Y.K.); nakanishi@yamadazaidan.jp (U.N.); 2Department of Veterinary Medicine, Research Center of Global Agromedicine, Obihiro University of Agriculture and Veterinary Medicine, Obihiro 080-8555, Japan; knabe@obihiro.ac.jp; 3Yamada Science Foundation, Osaka 544-8666, Japan

**Keywords:** Amyloid A amyloidosis, serum amyloid A, amyloid β, *Melissa officinalis*, microliter-scale high-throughput screening system, rosmarinic acid

## Abstract

Serum amyloid A (SAA) is one of the most important precursor amyloid proteins and plays a vital step in AA amyloidosis, although the underlying aggregation mechanism has not been elucidated. Since SAA aggregation is a key step in this pathogenesis, inhibitors are useful to prevent and treat AA amyloidosis, serving as tools to investigate the pathogenic mechanism. In this study, we showed that rosmarinic acid (RA), which is a well-known inhibitor of the aggregation of amyloid β (Aβ), displayed inhibitory activity against SAA aggregation in vitro using a microliter-scale high-throughput screening (MSHTS) system with quantum-dot nanoprobes. Therefore, we evaluated the amyloid aggregation inhibitory activity of blood and the deposition of SAA in organs by feeding mice with *Melissa officinalis* extract (ME) containing RA as an active substance. Interestingly, the inhibitory activity of ME-fed mice sera for SAA and Aβ aggregation, measured with the MSHTS system, was higher than that of the control group. The amount of amyloid deposition in the organs of ME-fed mice was lower than that in the control group, suggesting that the SAA aggregation inhibitory activity of serum is associated with SAA deposition. These results suggest that dietary intake of RA-containing ME enhanced amyloid aggregation inhibitory activity of blood and suppressed SAA deposition in organs. This study also demonstrated that the MSHTS system could be applied to in vitro screening and to monitor comprehensive activity of metabolized foods adsorbed by blood.

## 1. Introduction

Amyloidosis, which is characterized by the extracellular deposition of β-sheet-rich amyloid fibrils, is currently classified into more than 30 different types of medical associations based on precursor proteins or peptides [1,2,3]. Amyloid A (AA) amyloidosis is a protein-based disease frequently found in humans and livestock around the world [4,5,6]. It develops in patients with severe chronic inflammatory, infectious, and neoplastic diseases such as rheumatoid arthritis, ankylosing spondylitis, tuberculosis, and renal cell carcinoma, among others [7,8]. Amyloid deposits compress parenchymal cells and can lead to considerable dysfunction of the affected organs, cell degeneration, and even cancer in the body [9]. The fibrillar structure of amyloid fibrils, which also have roles in disease, may affect their ability to spread to different sites in a cell and between organisms in a prion-like mechanism [10]. However, the mechanism by which amyloid fibrils form in vivo and in vitro is largely unknown.

The major component of AA amyloid fibrils is serum amyloid A (SAA), which is an evolutionarily conserved family of acute-phase proteins in plasma. SAA has some biological functions that benefit the body, such as the regulation of inflammation and immunity, involvement in antibacterial and antiviral activities, and a role in lipid metabolism [11,12,13]. SAA is a major serum acute-phase protein which impacts 1% of patients with chronic inflammation such as rheumatoid arthritis and neoplastic diseases [14]. It is mainly produced in the liver but extra-hepatic synthesis involving the skin and adipose tissue has been reported [15]. In the clinical field, SAA is an important biomarker for inflammatory diseases [15,16,17]. The N-terminal of SAA is the most hydrophobic and amyloidogenic segment of the protein sequence and a driver of AA amyloidosis. Moreover, it is also crucial for the structure and stability of disease-associated AA amyloid fibrils [18,19,20]. Under normal physiological conditions, AA amyloid fibrils are harmless and soluble, but in inflammation or other damage-induced injuries (trauma, infection, tumors, etc.), the concentration of SAA rises sharply by about 1000 to 2000 times more than the normal level within 8 h [21,22,23]. Thus, an increased level of SAA can be used as an important biomarker of AA amyloidosis. Currently, anti-inflammatory drugs or colchicine are typically used in the therapy of AA amyloidosis to reduce SAA levels in blood circulation [10,18]. Although the mechanism of AA amyloidosis is currently unclear, the structure and basic composition of SAA have gradually become clear, including the use of cryo-electron microscopy (cryo-EM), allowing a better understanding of the pathogenesis [24,25]. As cryo-EM technology improves, this will provide a better understanding of how polymorphism is related to the disease phenotype and how fibril structure is affected by the cellular environment. At present, the 3D structure of human SAA1 protein, which became available in 2014 using cryo-EM, is a hexamer, with subunits displaying a unique four-helix bundle fold stabilized by a long C-terminal tail, similar to apolipoprotein E (apo E) [14,26].

Previously, we reported a real-time imaging method of amyloid β (Aβ_42_) and tau aggregation using quantum-dot (QD) nanoprobes and developed a microliter-scale high-throughput screening system (MSHTS) for aggregation inhibitors that applied this imaging method [27,28,29,30,31]. This screening system can be used to analyze a 5 µL sample volume when a 1536-well plate is used, and inhibitory activity can be estimated as half-maximal effective concentration (EC_50_). The major features of this method are that it can measure samples at a microliter scale and screen the activity of crude extracts containing various contaminants with high throughput (Appendix A). Using this method, we screened aggregation inhibitors from 52 spice extracts and revealed that rosmarinic acid (RA) is one active compound from summer savory (*Satureja hortensis*), which belongs to the *Lamiaceae* family [28]. The Aβ aggregation inhibitory activity of RA was clarified by Ono et al. [32]. It was subsequently reported that Aβ plaque deposition in RA-fed mice was reduced [33]. RA also shows high antioxidant activity, and antioxidants with a catechol moiety inhibited the aggregation of various amyloid proteins such as Aβ, tau, and α-synuclein [34,35,36,37].

To clarify the aggregation process of SAA and to inhibit Aβ aggregation, in this study, we used a QD imaging method and an interleukin 1 receptor antagonist knockout (IL-I raKO) mice model. IL-I raKO mice develop chronic arthritis that resembles human rheumatoid arthritis, and have about 10 times higher levels of SAA than control mice, making them useful for in vivo studies of AA amyloidosis [38]. First, we succeeded in real-time imaging of SAA aggregation using QDs and demonstrated that RA inhibits SAA aggregation in vitro using the MSHTS system by applying this imaging method. Then, after administering *Melissa officinalis* (lemon balm), which belongs to the *Lamiaceae* family, extract containing RA as an active substance in foods to mice, amyloidosis was compared between treatment and control groups. The results revealed that feeding *M. officinalis* extract (ME) increased the SAA aggregation inhibitory activity of serum in vitro and suppressed SAA deposition in the organs of mice. This result also suggests that the MSHTS system is capable of evaluating the activity of inhibition of amyloid aggregation in the body.

## 2. Results

### 2.1. Real-Time Imaging of SAA Aggregation Using QDs and Evaluation of the Aggregation Inhibitory Activity of RA by the MSHTS System

In this study, we used recombinant mouse SAA 1.1 (mSAA) protein to create an image of amyloid aggregation using QDs [36,39]. Long-term inflammation has this complication, and is best characterized by the SAA 1 genotype. We semi-quantified the amount of amyloid aggregates from the standard deviation (SD) of the brightness of each pixel in fluorescence microscopic images according to the methodology employed in previous studies [27,28,30,36]. In this method, the concentration of amyloid protein having a peak SD value is the optimum concentration for evaluating aggregation inhibitory activity. In this study, therefore, we measured SD values with varying concentrations of mSAA (Appendix A) and determined that about 50 µM was the optimal concentration to quantify inhibitory activity. A mixture of 49.3 µM mSAA and 30 nM QD605 in PBS was incubated at 37 °C in a 1536-well plate to induce aggregation, whose 2D images were captured by fluorescence microscopy at 0, 2, 4, 8, 16, 24, 48, and 72 h (Figure 1A). Aggregates were observed after 2 h of incubation, increasing over time (Figure 1A). Figure 1B plots the amount of aggregates estimated from the SD of the brightness of each pixel of a fluorescence microscope image [28,36]. Under similar conditions, Aβ began to aggregate at around 5 h [34], suggesting that the lag time of SAA aggregation is shorter than that of Aβ. Similar to Aβ, tau and α-synuclein [36], 3D images of SAA aggregates could be obtained by confocal laser microscopy.

Rosmarinic acid inhibited the aggregation of Aβ, tau, and α-synuclein, although the inhibitory activity was different [36]. Therefore, we evaluated whether RA inhibits the aggregation of SAA, which is a similar amyloidogenic protein, using the MSHTS system (Appendix A) that applies the QD imaging method (Figure 1D). To determine EC_50_ in this study, we used RA at various concentrations (final concentrations were 500, 100, 20, 4 and 0.8 μM) to inhibit mSAA aggregation. After incubation at 37 °C, aggregate images were captured by fluorescence microscopy, and the EC_50_ was calculated according to our previous reports [28,36]. The fluorescence micrographs showed that almost no aggregates were observed in the presence of 500 μM RA, as in the control (-RA), after 0 h incubation (Figure 1D). Even using confocal microscopy, almost no aggregates were observed in the presence of 500 μM RA (Figure 1D), compared to the control (Figure 1C). Separately, TEM results showed that when 500 μM of RA was incubated with SAA protein, the aggregation was significantly reduced (Appendix A). The EC_50_ value of RA calculated from the inhibition curve (Figure 1F) was 33.2 ± 2.19 μM. The EC_50_ value of RA for 30 μM Aβ was 20.6 μM, suggesting that RA has similar inhibitory activity against Aβ and SAA [36].

We also performed aggregate disruption experiments in the hope that RA would disrupt preformed SAA aggregates (Appendix A). However, RA was unable to destroy SAA once it had aggregated.

### 2.2. Inhibitory Activity of ME-Fed Mice Serum for Aggregation of mSAA and Aβ

Foods are digested and absorbed into the blood, which contains various components such as proteins, lipids, sugars, and inorganic salts. Since the MSHTS system is not easily affected by contaminants [27], in this study, we investigated whether an amyloid aggregation inhibitor taken in by the diet affects the aggregation inhibitory activity of blood. RA, which inhibits SAA aggregation in vitro (Figure 1D–F), is abundant in *Lamiaceae* plants. We reported that extracts from *Lamiaceae* plants showed high Aβ aggregation inhibitory activity, and that the main active component was RA [28]. Therefore, we administered a commercially available extract of a *Lamiaceae* plant, *Melissa officinalis*, which contains 1–2% RA and showed high Aβ aggregation inhibitory activity (EC_50_ = 0.00525 ± 0.00088 mg/mL, Appendix A). ME was administered to mice via their diet and we evaluated the activities of mSAA and Aβ aggregation in serum using the MSHTS system. In this study, we used IL-I raKO mice to induce AA amyloidosis, and amyloid deposition in organs was also examined, as described later. The blood of mice was collected from the group fed the ME (+ME) and the control group (−ME) to prepare serum. A total of 49.3 µM mSAA and 30 nM QD605 in PBS were incubated with various concentrations of serum (final concentrations—10%, 2%, 0.4%, 0.08%, 0.016% and 0.0032%) at 37 °C for 48 h in a 1536-well plate to induce aggregation and images were captured by fluorescence microscopy (Figure 2A). Interestingly, the sera of both −ME and +ME groups inhibited SAA aggregation activity in the presence of 10% serum. In the presence of 2% serum, slight protein aggregation was observed in the −ME group but no aggregation in the +ME group. EC_50_ values were calculated from the inhibition curves (Figure 2B). The serum EC_50_ value of the ME-treated group (0.299 ± 0.123%) indicated that inhibition activity was significantly higher than that of the control group (0.851 ± 0.193%).

To investigate the effect of feeding ME on serum, we also examined whether sera inhibited Aβ aggregation using the MSHTS system. The sera from both +ME and −ME groups inhibited Aβ aggregation (Figure 3A,B)—the EC_50_ value of the control group was 0.332 ± 0.025%, and that of the ME-treated group was 0.278 ± 0.036%. These results suggest that feeding ME significantly enhanced serum Aβ aggregation inhibitory activity (Figure 3C), as occurred with SAA.

### 2.3. Suppression of AA Amyloidosis by ME Feeding

IL-I raKO mice were intraperitoneally (i.p.). injected with 500 μL of murine amyloid enhancing factor (mAEF) and subcutaneously (s.c.). injected with 500 μL of 2% AgNO_3_ to induce AA amyloidosis. In this experiment, the IL-I raKO mice injected with mAEF induced amyloidosis and amyloid deposition began on day 2 [40]. AA amyloid deposits were also observed in other organs—adrenal gland, thyroid gland, and kidney. In this paper, we only evaluated the spleen, liver and gastrointestinal organs (Figure 4A). Staining with Congo red showed that amyloid was deposited around the follicles of the spleen (Appendix A). Murine AA amyloid extract was purified from the mice liver. A ~14 kDa mSAA band was detected by western blot analysis using anti-mouse SAA antibody (Figure 4B). Transmission electron microscopy (TEM) of the AA amyloid extract showed that the amyloid fibrils had a typical AA amyloid form with a width of 17.1 nm and a crossover distance of 112 nm (Figure 4C), although these values are somewhat different to previous data [26]. These results demonstrate that the above treatment indeed induced AA amyloidosis.

Next, we investigated whether ME treatment for 2 weeks suppressed AA amyloid deposition (Figure 4D). Immunohistochemistry (IHC) suggested that AA amyloid deposition was inhibited by ME treatment. We compared the +ME and −ME groups by measuring the areas of amyloid deposition in the liver and spleen to evaluate whether ME had a significant inhibitory effect on amyloid deposition. The results demonstrate that AA amyloid deposition was significantly suppressed by ME treatment (Figure 4D). Moreover, amyloid deposition was not observed with Congo red staining (data not show).

## 3. Discussion

SAA protein aggregates progressively and is deposited extracellularly, where it misfolds and evades normal clearance pathways, allowing the AA amyloidosis pathogenic process to ensue [41]. Amyloid fibrils are formed by the aggregation of monomeric protein precursors into fibrils by a common nucleation growth mechanism. This process includes three phases, nucleation, elongation and stationary, in which the fibrils are formed [10] (Figure 5). As fibrils grow, fragmentation can occur, yielding more fibrils ends that are capable of elongation by acting as a new aggregation-protein species [42,43]. Many similar forms of amyloidosis undergo this procedure, such as Alzheimer’s disease (AD), AL amyloidosis, and ATTR amyloidosis. Amyloid aggregates can disrupt cells by a variety of mechanisms, so precursors and toxic species that are formed cause cellular dysfunction and cell death [44,45]. Recent research suggested that AA amyloidosis could be transmitted by a prion-like infectious process through a seeding-nucleation mechanism [23,46]. However, the procedure is very sophisticated, and some reports suggested that pre-amyloid aggregates are the main cause of the induction of amyloidosis [47,48,49]. There is currently no effective treatment for amyloidosis by inhibiting the aggregation of amyloid fibrils.

To find an effective substance that inhibits Aβ aggregation, we previously screened 52 spices using the MSHTS system with QD nanoprobes, and revealed that RA abundant in *Lamiaceae* plants showed high inhibitory activity [28]. RA also suppressed Aβ deposition in the brain of AD mice [33]. At present, methods for detecting SAA aggregation mainly include radioimmunoassays, enzyme-linked immunosorbent assays, immune rate nephelometry, and others, but these methods are not suitable for high-throughput analysis. In this study, we thus attempted to extend the application of the MSHTS system to screen for SAA aggregation inhibitors. We succeeded in real-time imaging of the SAA aggregation process by QDs and quantified aggregation inhibition by RA in vitro (Figure 1). Interestingly, the MSHTS system also detected enhanced SAA and Aβ aggregation inhibitory activity in the serum of mice that ingested ME abundant in RA (Figure 2 and Figure 3). Dietary intake of ME suppressed SAA deposition in organs, suggesting that RA suppresses SAA aggregation not only in vitro but also in vivo (Figure 4). These results suggest that the MSHTS system can be used to evaluate amyloid aggregation inhibitory activity in vivo as well as in vitro.

Why was the SAA aggregation inhibitory activity in serum enhanced by the RA diet (Figure 2)? There are two possible causes for the enhancement of SAA aggregation inhibitory activity in serum—a decrease in SAA concentration in the assay system and/or an increase in the aggregation inhibitor. Our recent report showed that the SAA concentration in mouse serum induced by the same method as in this paper was 0.1 mg/mL (~7 µM) after 2 weeks [40]. Even if ME had the effect of lowering SAA concentration in serum, the ratio was very small compared to 49.3 μM, the concentration of SAA used in the experiment (about 14% when 7 μM was reduced to 0 μM). Therefore, we consider that it is difficult to explain the 2.8-fold difference in EC_50_ value only by the circulating SAA found in serum. Hase et al. [50] reported that the RA concentration of plasma in mice fed 0.5% RA for 7 weeks was only ~1 µM. Since the EC_50_ value of RA against 49.3 μM SAA was 33.2 μM (Figure 1), it is not only intact RA absorbed into the blood by the diet that is involved in the enhancement of serum aggregation inhibitory activity. Hase et al. also reported that RA suppresses Aβ aggregation by increasing monoamine secretion [50]. We speculate that intact RA, RA metabolites, and some biological substances affected by RA may influence the SAA aggregation inhibitory activity of serum. These studies require detailed analyses of pharmacokinetics/dynamics/metabolism, which we will consider as a follow-up research topic.

Based on our in vitro and in vivo studies, we speculate that when RA in ME is fed to mice for 2 weeks, it is absorbed through the alimentary system, entering blood circulation and flowing to various organs throughout the body. Due to the presence of RA and/or metabolized RA in blood circulated throughout the body, this could inhibit incorrect folding, causing the deposition of fibrils in organs. The mechanism of inhibition might include stabilizers of the native conformation of amyloidogenic precursors, inhibitors of fibrils, amyloid fibril disruptors, and promoters of amyloid clearance [51]. Surprisingly, the results of the MSHTS system revealed that serum of −ME group mice also displayed SAA inhibition activity (Figure 2) and Aβ aggregation (Figure 3). The reason why serum of −ME group mice exhibited amyloid aggregation inhibitory activity is unknown, but one possibility is that these mice contain anti-amyloid antibody. Antibodies can eliminate and clear visceral and brain amyloid deposits in mice [52,53,54]. Moreover, various components in serum may simply compete with the process of aggregation and inhibit amyloid formation. Independent of the mechanism, the inhibitory activity of +ME serum was higher than that of −ME serum. This indicates that the serum from the +ME treatment group mice that were fed ME, in addition to anti-amyloid antibodies and other components, also contains an effective active ingredient, ME. The main function of anti-amyloid therapy is to promptly reduce or eliminate amyloid precursors. Other reports on amyloidosis therapy showed a favorable outcome for AA amyloidosis when SAA concentration was maintained below 10 mg/L in the human body [51].

There are many other proteins such as Aβ, tau, and α-synuclein which can induce different types of diseases caused by protein misfolding [55,56]. The toxicity of soluble oligomers is a trigger for neurodegenerative disorders such as AD and Parkinson’s disease although, by targeting distinct species of Aβ and tau for therapeutic intervention, there is currently no effective way to completely cure these diseases [57,58]. For amyloidosis, the preferred drug therapy and immunotherapy were combined with oral cyclophosphamide, bortezomib and dexamethasone [59,60,61]. However, therapies that directly target amyloid deposits in organs to clear them have thus far been unsuccessful. In this study, we elucidated that RA- and/or RA-rich ME can inhibit mSAA aggregation in vivo and in vitro. Currently there is no data that show the use of serum treated by RA for the treatment of AA amyloidosis. We also demonstrated that serum treated by RA could be useful in the design of an inhibitor in drug discovery for the prevention and early treatment of amyloidosis. This novel therapy to eliminate amyloid aggregation would be applicable to all forms of amyloidosis, and delay or even halt its progression.

In this paper, we demonstrated that the MSHTS system could evaluate amyloid aggregation inhibitory activity not only in vitro but also in vivo. The MSHTS system has also been successfully automated [27]. It is expected that this system can be used in the future for high-throughput evaluation of aggregation inhibitors after absorption into the living body. This may lead to a dramatic progress in the search for substances that are truly effective in the living body.

## 4. Materials and Methods

### 4.1. Mice

In this study, all procedures used 13-month-old IL-I raKO mice of randomly mixed gender because there is no difference in the prevalence and severity of experimental AA amyloidosis in male and female mice [38]. Their use was approved by the Research Center of Global Agromedicine of Obihiro University of Agriculture and Veterinary Medicine, and adequate steps were taken to ensure that animals did not suffer unnecessarily at any stage of an experiment, whether acute or chronic. The physiologic features and genetic characteristics of IL-I raKO mice have already been described [38,62]. Three or four mice were housed in a single cage and they had free access to water and food (CE-2, CLEA, Tokyo, Japan). The animal experimentation was conducted according to the protocol reviewed and approved by (OUAVM) Animal Care and Use Committee (Permission No. 19-179: 7-10-2019).

### 4.2. Preparation of Mouse SAA Protein, AEF and Amyloid Extracts

Murine AEF and total amyloid extracts were prepared from the liver of mice with AA amyloidosis as previously described [38,63]. Briefly, 1 g of liver tissue was homogenized (TAITEC, Tokyo, Japan) with 10 mL of PBS buffer and placed on ice for 30 s. AA amyloid extracts were purified from homogenized liver using a modified Pras et al. method [63]. In brief, 1 g of tissue was homogenized with 10 mL of saline and ultra-centrifuged (Beckman Coulter Optima L-100 XP Ultracentrifuge, Backman, Brea, CA, USA) at 35,000 *g* for 50 min. The pellet was resuspended with saline, homogenized and ultra-centrifuged. The pellet was resuspended in saline, the previous process was repeated, and after centrifuging a total of 10 times to better remove impurities other than fibrils, the pellet was resuspended in distilled water. The sample was ultra-centrifuged at 35,000 *g* for 45 min, and the supernatant was collected. The supernatant was centrifuged at 125,000 *g* for 1 h, then the pellet was stored as an amyloid extract. The AEF and amyloid extracts were stored at −80 °C until use.

The preparation of mSAA (SAA1.1: accession No. NP_033143) was performed as described previously with some modifications. Briefly, mSAA cDNA was inserted into the pET-15b vector, expressed in Rosetta *Escherichia coli* and purified [39]. *E. coli* was grown in Luria-Bertani medium supplemented with 25 µg/mL chloramphenicol and 100 µg/mL ampicillin at 37 °C. When OD_600_ of the culture reached 0.6, cells were induced by 1 mM isopropyl thio-β-D-galactoside for 5 h. After centrifugation at 5000 rpm, the pellet was suspended in 10 mM HEPES buffer (pH 7.4) and washed twice. After washing, the pellet was resuspended in buffer A (10 mM HEPES, pH 7.4, 500 mM NaCl, 0.5% Triton100), sonicated (5 kHz) for 5 min and centrifuged at 18,000 rpm for 30 min. The supernatant was applied to a Ni Sepharose^TM^ 6 Fast Flow column (GE Healthcare Life Sciences, Marlborough, MA, USA) then washed with buffer B (10 mM HEPES, 500 mM NaCl, pH 7.4, 0.5% Triton100, 50 mM imidazole). The mSAA protein was eluted using buffer C (10 mM HEPES, pH 7.4, 100 mM NaCl, 500 mM imidazole). mSAA protein was dialyzed for 8 h in 10 mM HEPES buffer then centrifuged at 15,000 rpm for 30 min at 4 °C. The Lowry [64] method was used to determine mSAA protein concentration in the supernatant. The prepared mSAA was diluted with phosphate-buffered saline (PBS; pH 7.4) at a concentration of 1 mg/mL and stored at −80 °C until use.

### 4.3. Induction of AA Amyloidosis Using AEF

Ten IL-I raKO mice were divided into two groups (+ME and −ME groups, +ME *n* = 5 in separate experiments; −ME *n* = 5 in separate experiments) and each mouse was i.p. injected with 500 μL of AEF and s.c. injected with 500 μL of 2% AgNO_3_ to induce AA amyloidosis. The ME-treated group (+ME) was fed 1% *Melissa officinalis* extract which contained 1–2% RA (lemon balm extract powder ME, Maruzen Pharmaceuticals Co., Ltd., Japan) in a mixed diet starting 3 days (−3 day) before amyloidosis was induced. The control group (−ME) was fed a normal diet. After feeding for 2 weeks, mice were euthanized under deep isoflurane anesthesia, and all blood in the heart was collected. The liver, spleen and ileocecal junction were obtained by necropsy and fixed in 10% neutral buffered formalin. All samples, after collecting blood, were transfer into a new 1.5 mL tube and centrifuged at 5000 rpm for 15 min at room temperature. The collected supernatant (serum) was stored at −80 °C until use.

### 4.4. Histopathology

Three serial sections from all mice were made from formalin-fixed and paraffin-embedded tissues then stained with hematoxylin and eosin (HE), Puchtler’s alkaline Congo red (Congo red) and assessed by IHC. IHC was performed using goat anti–mouse SAA1.1 polyclonal antibody (1:400, R&D Systems, Minneapolis, MN, USA) using the Envision system (Agilent, Santa Clara, CA, USA) as described previously [40]. AA amyloid deposits were identified by a Congo red-positive homogenous matrix with apple-green birefringence under polarized light, and tested positive for mouse SAA1.1 by IHC.

### 4.5. Image Analysis of Amyloid Deposition

After IHC staining, 20 images of the liver and spleen in each mouse were randomly taken at ×200 magnification (Appendix A). The severity of amyloid deposit was determined as a percentage of the immunopositive area relative to the total area. The immunopositive area was measured using Image J software v.1.53b (National Institutes of Health, Bethesda, MD, USA). The data obtained in image analysis were statistically analyzed by Welch’s *t*-test.

### 4.6. Western Blot Analysis of Total Amyloid Extracts

The total amyloid extracts were mixed with 4× Laemmli sample buffer (Bio-Rad, Hercules, CA, USA) and boiled at 100 °C for 5 min. Samples and molecular weight marker (ATTO, Tokyo, Japan) were separated by sodium dodecyl sulfate-polyacrylamide gel electrophoresis (SDS-PAGE) on a 15% polyacrylamide gel (Bio-Rad) [65]. Thereafter, SAA protein was transferred onto a polyvinylidene membrane (PVDF), and blocked with 1% skimmed milk in PBS with Tween−20 for 1 h at room temperature. The primary antibody was anti-mouse SAA1 goat polyclonal (R&D Systems), the secondary antibody was horseradish peroxidase (HRP)-conjugated donkey anti-goat IgG (Santa Cruz Biotechnology, Dallas, TX, USA). After staining in Clarity^TM^ Western CEL substrate (Bio-Rad), bands were detected by ImageQuant LAS 500 (GE Healthcare Life Sciences).

### 4.7. Transmission Electron Microscopy Observations of Amyloid Fibrils

Ten µL of AA amyloid extract and aggregated mSAA with or without RA at 37 °C for 48 h was applied on parafilm and absorbed to corodion membrane-coated 150-mesh copper grids (Nisshin EM, Tokyo, Japan) for 10 min, negatively stained with 1% phosphotungstic acid for 5 min twice, then washed with PBS at room temperature. Specimens were detected under an H-7600 transmission electron microscope (Hitachi, Tokyo, Japan) at 80 kV.

### 4.8. Imaging of mSAA Aggregation

mSAA samples for fluorescence microscopic observation were prepared as follows—49.3 µM mSAA, 30 nM QD605 (Q21501MP, Thermo Fisher Scientific, Waltham, MA USA) in PBS. Five μL of sample was transferred to each well of a 1536-well plate and the plate was centrifuged (PlateSpin, Kubota, Tokyo, Japan) at 3700 rpm for 5 min at room temperature. After centrifugation, the plate was incubated at 37 °C in an air incubator (SIB-35, Sansyo, Tokyo, Japan), and observed at 0, 2, 4, 8, 16, 24, 48, and 72 h using an inverted fluorescence microscope (TE2000, Nikon, Tokyo, Japan) or a confocal laser microscope (Nikon C2 Plus, Nikon). To estimate the amount of aggregates from a 2D image, the SD values of fluorescent intensity of each pixel (200 × 200 pixels) in the central region of fluorescence micrographs were measured by ImageJ software Ver 1.53b (National Institute of Health, Bethesda, MA, USA) according to our previous reports [27,28,36].

### 4.9. Measurement of Amyloid Aggregation Inhibitory Activity of RA or Sera Using MSHTS System

Various concentrations of RA or serum were mixed with 49.3 µM mSAA or 30 µM Aβ (human amyloid peptide of Aβ_42_ 4349-v) and 30 nM QD605 in PBS. Five μL of sample was transferred into a 1536-well plate, the plate was centrifuged at 3700 rpm for 5 min, then incubated at 37 °C for 48 h in an air incubator (SIB-35). Images of mSAA samples were captured at 48 h, and of Aβ samples at 24 h. To determine EC_50_ values, the images of each well were observed by an inverted fluorescence microscope (TE2000) using a 4× objective equipped with a color CCD camera (DP72, Olympus, Tokyo, Japan) according to our previous reports [28,36]. Briefly, the fluorescence intensities of 200 × 200 pixels in the central region of each well were assessed, and the RGB value was adjusted to 50% of the original value. SD values were measured by ImageJ software. EC_50_ values for inhibition of Aβ aggregation were determined according to our previous reports [27,28,36].

### 4.10. RA Post-Aggregation Inhibition Experiment

A sample solution of 49.3 µM mSAA and 30 nM QD in PBS was prepared and 5 µL was injected into a 1536-well plate. After centrifugation of the plate, it was incubated at 37 °C for 48 h. One µL of 1000 µM RA, giving a final concentration of 167 µM, was added to each aggregation sample, and the plate was centrifuged. The plate was incubated for 120 h, and images were taken every 24 h until 120 h. Images were analyzed using ImageJ software to calculate SD values.

## Figures and Tables

**Figure 1 ijms-21-06031-f001:**
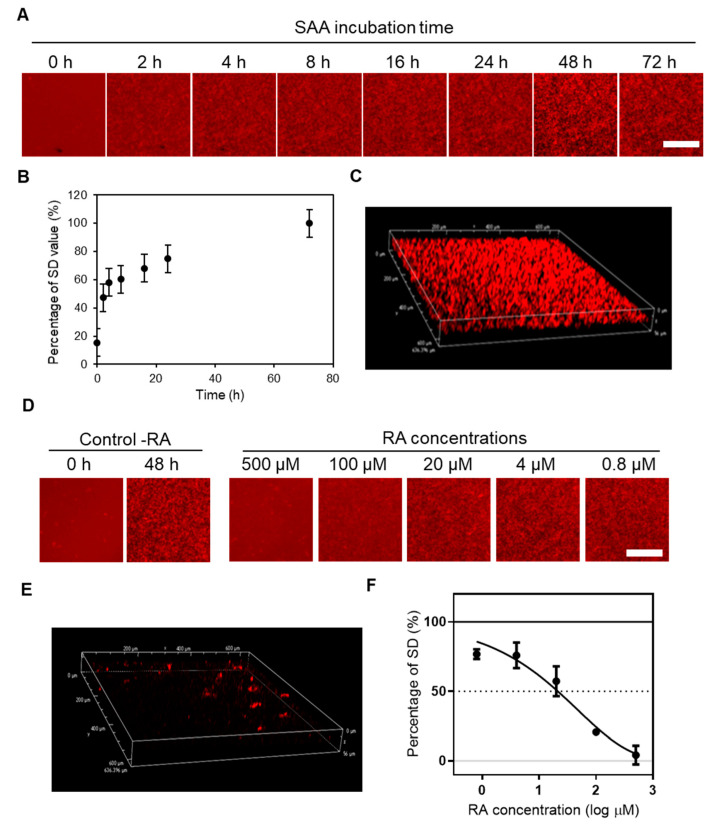
Real-time imaging of mouse serum amyloid A (mSAA) aggregation using quantum-dots (QDs) and evaluation of inhibitory activity of rosmarinic acid (RA) using the microliter-scale high-throughput screening system (MSHTS) system. (**A**) Real-time 2D imaging of mSAA aggregations were captured by conventional fluorescence microscopy using a 4× objective for 0, 2, 4, 8, 16, 24, 48, and 72 h. The images show the central area in a well observed by fluorescence microscopy at each incubation time. (**B**) Standard deviation (SD) value of each image by time-dependent mSAA protein aggregation. (**C**) 3D imaging of mSAA aggregation was captured by confocal microscopy. (**D**) 2D fluorescence microscopic images of mSAA aggregates in the absence (control) or presence of various concentrations of RA after 48 h incubation. (**E**) 3D visualization of mSAA aggregation in the presence of 125 µM RA. (**F**) Estimation of EC_50_ values from an inhibition curve (*n* = 3 separate experiments). Scale bars in fluorescent micrographs indicate 100 μm. Each plot represents mean ± SEM. The percentage of SD was defined as SD values before and after the incubation of samples (0% and 100%, respectively).

**Figure 2 ijms-21-06031-f002:**
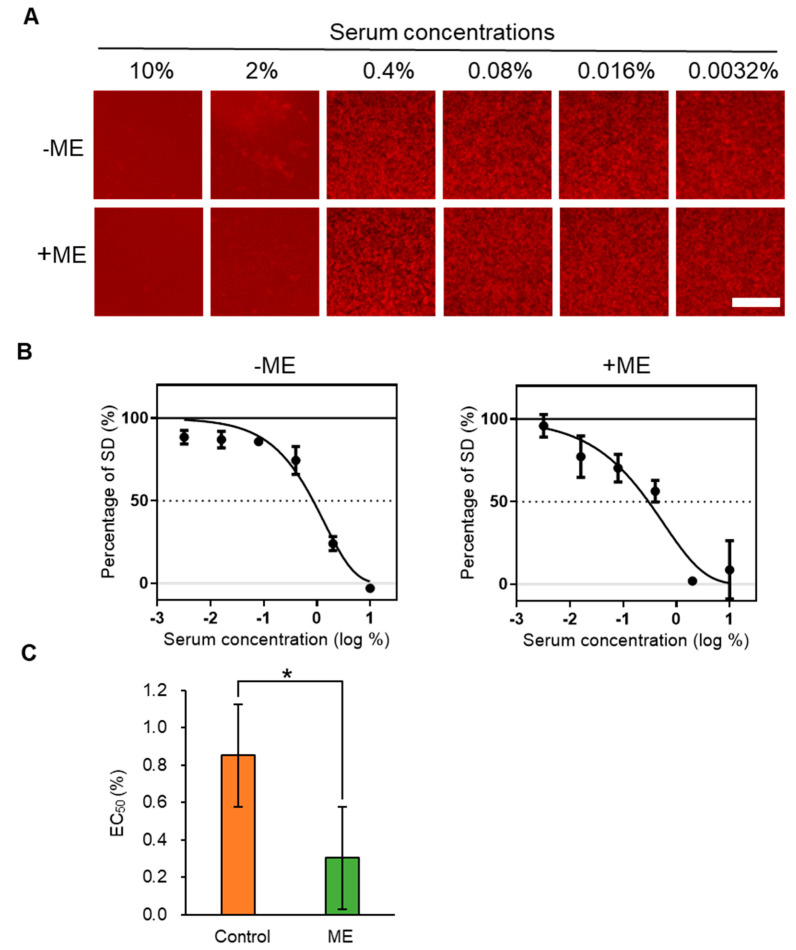
Estimation of EC_50_ of serum for mSAA aggregation by the MSHTS system. (**A**) mSAA containing QD nanoprobes were incubated with various concentrations of +ME or −ME group sera. (**B**) Estimation of EC_50_ values from inhibition curves. (**C**) Comparison of the EC_50_ values of +ME and −ME groups (* *p* < 0.05, Welch’s *t*-test) (*n* = 3 separate experiments). Each plot and bar graphs represents mean ± SEM.

**Figure 3 ijms-21-06031-f003:**
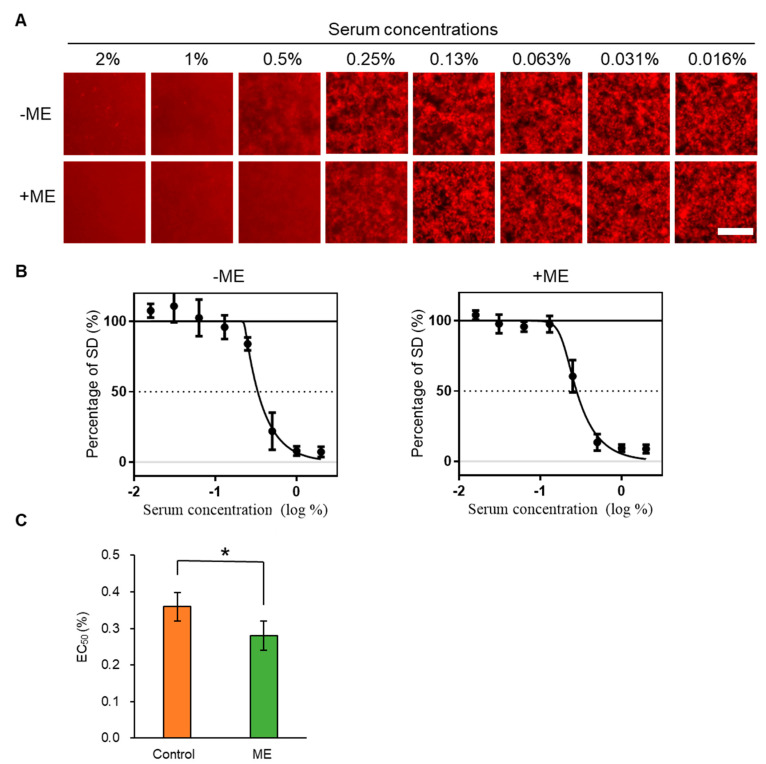
Estimation of EC_50_ of serum for amyloid beta (Aβ) aggregation by the MSHTS system. (**A**) Aβ-containing QD nanoprobes were incubated with various concentrations of +ME or −ME group sera. (**B**) Estimation of EC_50_ values from inhibition curves. (**C**) Comparison of the EC_50_ values between +ME and −ME groups (* *p* < 0.05, Welch’s *t*-test) (*n* = 3 separate experiments). Each plot and bar graphs represents mean ± SEM.

**Figure 4 ijms-21-06031-f004:**
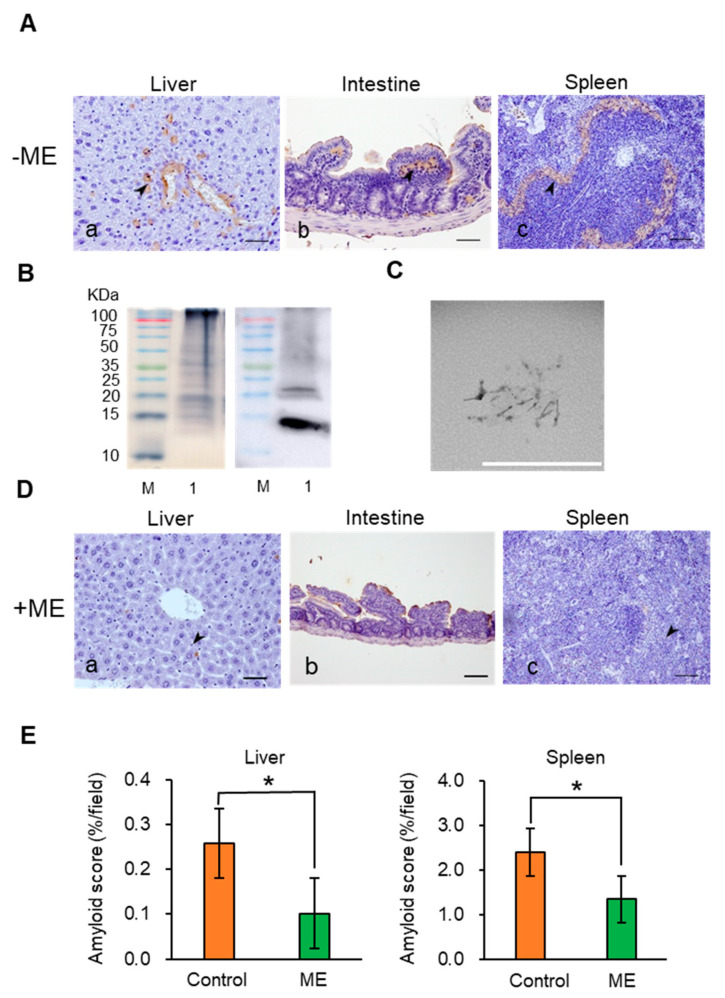
Amyloid deposition in mouse organs with or without RA treatment. (**A**) Experimentally induced AA amyloidosis detected by IHC staining (−ME group): (a) Liver. AA deposit was observed in the central vein and in the space of Disse (arrowhead) (*n* = 5 separate experiments). (b) Intestine. AA deposit was observed in the lamina propria (arrowhead). (c) Spleen. AA deposit was observed adjacent to the white pulp (marginal zone) (arrowhead). Scale bars indicate 100 μm. (**B**) SDS-PAGE (left) and Western Blot (right) of murine AA amyloid extracts. M, molecular weight marker; Lane 1: full murine AA amyloid extracts. (**C**) Transmission electron microscopic observation of amyloid fibrils. Scale bar indicates 1 µm. (**D**) Induced AA amyloidosis of ME-treated group (+ME group) mice (IHS staining). (a) Liver. Slight AA deposit was observed (arrowhead). (b) Intestine. AA deposit was not observed. (c) Spleen. Slight AA deposit was observed (arrowhead) (*n* = 5 separate experiments). Scale bars indicate 100 μm. (**E**) Amyloid deposition analysis in liver and spleen with +ME and control group (* *p* < 0.05, Welch’s *t*-test). Each error bar represents mean ± SEM.

**Figure 5 ijms-21-06031-f005:**
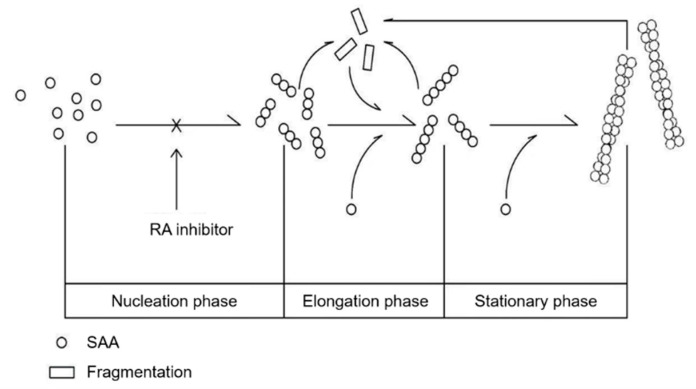
Models showing amyloid fibril formation and inhibition by RA. The mSAA oligomer can assemble further to form higher order oligomers, then rapidly recruit other monomers from the surroundings to assemble amyloid fibrils. RA can inhibit an oligomer from forming a higher order oligomer or/and amyloid fibrils in nucleation and elongation phases, but cannot degrade the formed fibrils.

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
