# Peer review of "Dietary Intake of Rosmarinic Acid Increases Serum Inhibitory Activity in Amyloid A Aggregation and Suppresses Deposition in the Organs of Mice"

_ijms, 2020, doi:10.3390/ijms21176031_

Round 1

Reviewer 1 Report

The paper “Dietary intake of rosmarinic acid increased serum  inhibitory activity in amyloid A aggregation and suppressed deposition in organs” by  Xuguang Lin et al. is focused on the effect of the rosmarinic acid (RA) in the aggregation of serum amyloid a (SAA) protein. The author demonstrates the inhibitory effect on SAA aggregation by RA in mice fed with Melissa officinalis extract by using microliter-scale high-throughput screening (MSHTS) system with quantum-dot nanoprobes. Experiments were well designed and performed. Conclusions are well supported by data. I recommend the paper for publication in International journal of molecular science.

Author Response

Thank you for your positive evaluation. The paper was further revised and English proofreading was performed according to the opinions and requests of other reviewers.

Reviewer 2 Report

In this manuscript, Lin et al. evaluate the anti-amyloidogenic properties of rosmarinic acid in context of AA amyloidosis.  They use in vitro methods to evaluate the abilities of RA and blood from mice fed an RA-enriched diet to inhibit SAA aggregation.  They further attempt to evaluate the efficacy of RA against AA amyloidosis in vivo.  In general, the manuscript is clearly written but there are some problems with the methods and approach that should be addressed and are detailed below as major and minor comments:

Major comments: 

1)  Images in Figure 4 and Figure S4:  It is not clear what is being stained here- the legend states AA amyloidosis detected by IHC staining.  Methods state that congo red/SAA IHC dual staining was used, but congo red staining indicative of amyloidosis is not apparent in these figures.  It needs to be shown because what is being shown in the figures isn't really convincing that there are increases in AA amyloid deposits.  This is also in part due to small images of moderate quality.

2)  The interpretation of results from Figure 2 is a little confusing.  What does ME feeding do to concentrations of circulating SAA, and are RA levels increased in serum with ME feeding?

Minor comments:

P2, lines 48-49 "Notably, ... inflammation". What is meant by this sentence?  Are you trying to say that 1% of patients with chronic inflammation develop AA amyloidosis?

P2, lines 74-75 "and revelaed... family" Rephrase this sentence to better convey the point.

P2, lines 80-81:  It should be further explained here why IL-1 raKO mice are used, with reference(s) for this mouse as an AA amyloidosis model.

P2, line 92:  Recombinant mouse SAA should be more clearly defined.  Is this SAA1 or SAA2 and what form (following nomenclature assigned by Sipe PMID 10211414)?  Accession number should be included in methods for clarity.

P2 line 93:  Provide rationale for using 49.3uM mSAA in the aggregation studies.

P3 lines 111-112 redundant provision of RA EC50 for SAA experiements

P4 lines 147-149:  Provide serum EC50 units (%?).  Note that an EC50 of .299 was incorrectly stated to be higher than .851.

P6 line 168:  define mAEF upon first use (it is not defined until methods)

P6 line 172:  Purification of AA extract is unclear- was this a standard tissue extraction or an extraction like guanidine-HCl that is meant to solubilize amyloid deposits?

P line 231:  "enhanced the inhibitory activity of serum SAA aggregation"- I don't think this exactly conveys what is meant.

Some background on in vivo pharmacokinetics/dynamics/metabolism of ME/RA could be provided in the discussion

Methods:

What was the sex of the mice?

Lines 308-313:  Interchanging ME and RA groups is confusing.  It is also confusing when this is done in the figures and legends- it would be more clear/precise to stick with ME to indicate treatments since the reader is aware that ME is a source of RA.

Lines 319-327:  The staining procedures described here aren't apparent in the figures/legends.  

Line 335:  How was the total amyloid extract prepared?

There are minor grammatical errors throughout

Author Response

In this manuscript, Lin et al. evaluate the anti-amyloidogenic properties of rosmarinic acid in context of AA amyloidosis.  They use in vitro methods to evaluate the abilities of RA and blood from mice fed an RA-enriched diet to inhibit SAA aggregation.  They further attempt to evaluate the efficacy of RA against AA amyloidosis in vivo.  In general, the manuscript is clearly written but there are some problems with the methods and approach that should be addressed and are detailed below as major and minor comments:

Thank you for thoroughly reviewing our methodology and results. We have carefully considered your comments and made revisions where necessary.

Major comments:

1)  Images in Figure 4 and Figure S4:  It is not clear what is being stained here- the legend states AA amyloidosis detected by IHC staining.  Methods state that congo red/SAA IHC dual staining was used, but congo red staining indicative of amyloidosis is not apparent in these figures.  It needs to be shown because what is being shown in the figures isn't really convincing that there are increases in AA amyloid deposits.  This is also in part due to small images of moderate quality.

Following the reviewer's suggestions, we increased the size of images in Figure 4 and Figure S4 (revised Figure S7) to improve image quality. Furthermore, we added a Congo red stain image (revised Figure S6), and modified the relevant text in section 2.3.

2)  The interpretation of results from Figure 2 is a little confusing.  What does ME feeding do to concentrations of circulating SAA, and are RA levels increased in serum with ME feeding?

Our recent report showed that SAA concentration in mouse serum induced by the same method as was used in this paper was 0.1 mg/ml (~ 7 µM) after 2 weeks (Watanabe et al., 2017; Ref. 40). Even if ME (or RA) had the effect of lowering SAA concentration in serum, the ratio was very small compared to the concentration of SAA (49.3 μM) used in the experiment (about 14% when 7 μM was reduced to 0 μM). Therefore, we consider that it is difficult to explain the 2.8-fold difference in EC50 value only by the circulating SAA that exists in serum. Hase et al. (2019, Ref 51) reported that RA concentration of plasma in mice fed with 0.5% RA for 7 weeks was only ~1 µM. Since the EC50 value of RA against 49.3 μM SAA was 33.2 μM (Fig. 1), it is not only intact RA taken into the blood by the diet that is involved in the enhancement of serum aggregation inhibitory activity. We speculate that RA metabolites and/or some biological substances also affected by RA may influence the SAA aggregation inhibitory activity of serum. We added the above explanation in the discussion (P10, lines 260-277).

Minor comments:

P2, lines 48-49 "Notably, ... inflammation". What is meant by this sentence?  Are you trying to say that 1% of patients with chronic inflammation develop AA amyloidosis?

We revised the description according to the reviewers' comment (P2, lines 56-57).

P2, lines 74-75 "and revelaed... family" Rephrase this sentence to better convey the point.

We revised the description according to the reviewers' request (P2, lines 82-87).

P2, lines 80-81:  It should be further explained here why IL-1 raKO mice are used, with reference(s) for this mouse as an AA amyloidosis model.

Following the reviewer's suggestion, we added a description (P3, lines 93-95).

P2, line 92:  Recombinant mouse SAA should be more clearly defined.  Is this SAA1 or SAA2 and what form (following nomenclature assigned by Sipe PMID 10211414)?  Accession number should be included in methods for clarity.

We used recombinant mouse SAA 1.1 (accession No. NP_033143) in this study.

According to the reviewer's comment, information of recombinant mouse SAA (P3, lines 107-109) and accession No. (P12, line 347) were added.

P2 line 93:  Provide rationale for using 49.3uM mSAA in the aggregation studies.

We semi-quantified the amount of amyloid aggregates from the standard deviation (SD) of the brightness of each pixel in fluorescence microscopic images according to a previous paper. At that time, the concentration of amyloid protein having a peak SD value is the optimum concentration for evaluating aggregation inhibitory activity. In this study, therefore, we measured SD values with varying concentrations of mSAA (Figure S2) and determined that 50 µM was the optimal concentration to quantify the inhibitory activity. Since SAA aggregates were often observed at about 1 mg/ml (~70 µM) (e.g., Ref. 16), we believe that our experimental conditions are comparable to those of other reports. We added a description about the mSAA concentration (P3, lines 109-115).

P3 lines 111-112 redundant provision of RA EC50 for SAA experiements

We shortened the explanation according to the reviewer's comments (P4, lines 137-139).

P4 lines 147-149:  Provide serum EC50 units (%?).  Note that an EC50 of .299 was incorrectly stated to be higher than .851.

As the reviewer pointed out, our original explanation was incorrect. We corrected it with an accurate explanation (P5, lines 175-177).

P6 line 168:  define mAEF upon first use (it is not defined until methods)

Following the reviewers' comment, we defined mAEF upon first use (P7, lines 196-197).

P6 line 172:  Purification of AA extract is unclear- was this a standard tissue extraction or an extraction like guanidine-HCl that is meant to solubilize amyloid deposits?

The AA extract used in this study, which was purified from mouse liver, was a soluble fraction of AA amyloid deposits. Detailed methods are described in section 4.2.

P 7line 231:  "enhanced the inhibitory activity of serum SAA aggregation"- I don't think this exactly conveys what is meant.

We deleted this sentence as it is redundant.

Some background on in vivo pharmacokinetics/dynamics/metabolism of ME/RA could be provided in the discussion

Following the reviewer's suggestion, we added a description to the discussion (P10, lines 268-277).

Methods:

What was the sex of the mice?

We added the gender of the mice (P11, lines 326-328). In a previous study and in a preliminary experiment, there were no differences in the prevalence and severity of experimental AA amyloidosis between male and female mice.

Lines 308-313:  Interchanging ME and RA groups is confusing.  It is also confusing when this is done in the figures and legends- it would be more clear/precise to stick with ME to indicate treatments since the reader is aware that ME is a source of RA.

The group name was unified to ME according to the reviewer's comments (P12, lines 365-366).

Lines 319-327:  The staining procedures described here aren't apparent in the figures/legends. 

We modified the Methods section and added an image of Congo red staining to the Supplementary information (Figure S6).

Line 335:  How was the total amyloid extract prepared?

According to the reviewer's request, we added an explanation (P12, lines 335-337).

There are minor grammatical errors throughout

The manuscript was thoroughly proofread by a native English speaker and professional editing service.

Reviewer 3 Report

This is an interesting and novel study. However, there are some major concerns that should be addressed before publication.

1) The English grammar of the manuscript is very poor and needs to be thoroughly corrected. In the attached version of the manuscript, notes have been inserted with suggestions of how the text can be rephrase. However, the whole manuscript needs to be proofread.

2) Figure 1a. The aggregates formed need to be characterized by TEM or Th T binding etc. to verify the formation of amyloid fibrils.

3) Figure 1b. The label of the Y-axis must be clarified. This applies to all graphs that show the percentage of SD value.

4) Figure 1d. The inhibition of fibril formation by RA should be verified in parallel with other methods such as TEM and/or Th T bindning

5) Line 123. The images for the control are shown at 0 h and 48 h. Why are the images at various concentrations of RA taken after 72 h?

6) In materials and methods a description of how the Aβ samples were prepared is lacking. In addition a description of which Aβ variant that was used in the study is lacking. 

7) At line 147 it is stated that "The serum EC50 value of the ME-treated group (0.299 ± 0.123%) was significantly higher than that of the control group (0.851 ± 0.193%)". From these numbers it appears to me that the EC50 value of the ME-treated group is lower than that of the control group. This confusions need to be clarified.

8) Line 153 should say ME-treated instead of RA-treated (see note in the attached file).

9) Figure 4b. This experiment needs a control sample from control mice.

Author Response

This is an interesting and novel study. However, there are some major concerns that should be addressed before publication.

1) The English grammar of the manuscript is very poor and needs to be thoroughly corrected. In the attached version of the manuscript, notes have been inserted with suggestions of how the text can be rephrase. However, the whole manuscript needs to be proofread.

Thank you for your suggestions. We revised the manuscript based on the attached version of the manuscript, including your suggestions. The manuscript was also thoroughly proofread by a native English speaker and professional editing service.

2) Figure 1a. The aggregates formed need to be characterized by TEM or Th T binding etc. to verify the formation of amyloid fibrils.

According to the reviewers' comment, we added TEM images of mSAA aggregates with QD to Figure S3 (P4, lines 134-136).

3) Figure 1b. The label of the Y-axis must be clarified. This applies to all graphs that show the percentage of SD value.

The percentage of SD value is correlated with the amount of amyloid aggregates (Ishigaki et al., 2013). According to the reviewer's comment, an explanation of the Y-axis was specified in the figure legend (P5, lines 153-154).

4) Figure 1d. The inhibition of fibril formation by RA should be verified in parallel with other methods such as TEM and/or Th T bindning

According to the reviewers’ request, we added TEM images of mSAA with or without RA in Figure S3 (P4, lines 134-136).

5) Line 123. The images for the control are shown at 0 h and 48 h. Why are the images at various concentrations of RA taken after 72 h?

We made a mistake in the description of the figure legend. Correctly, Fig. 4D shows the results of 48 h incubation. We revised the figure legend accordingly (P5, line 150).

6) In materials and methods a description of how the Aβ samples were prepared is lacking. In addition a description of which Aβ variant that was used in the study is lacking.

In this study, we used commercially available human Aβ42 peptide. Following the reviewer's request, we added detailed information about the experiment on Aβ (P14, lines 421-425).

7) At line 147 it is stated that "The serum EC50 value of the ME-treated group (0.299 ± 0.123%) was significantly higher than that of the control group (0.851 ± 0.193%)". From these numbers it appears to me that the EC50 value of the ME-treated group is lower than that of the control group. This confusions need to be clarified.

As pointed out by reviewer #2, our original explanation was incorrect. We corrected it accordingly (P5, lines 175-177).

8) Line 153 should say ME-treated instead of RA-treated (see note in the attached file).

According to the reviewers' comment, "RA-treated" was changed to "ME-treated" (P5, line 181).

9) Figure 4b. This experiment needs a control sample from control mice.

The mouse model of AA-amyloidosis used in this study has already been well studied (Ref. 38 and 40), so we determined that a control would not be necessary.

Round 2

Reviewer 3 Report

The authors have responded adequately to the comments and corrected the English language. I am happy with these answers and consider the manuscript publishable